# The Prevalence of *Salmonella* and *Campylobacter* on Broiler Meat at Different Stages of Commercial Poultry Processing

**DOI:** 10.3390/ani12182460

**Published:** 2022-09-17

**Authors:** Hudson T. Thames, Courtney A. Fancher, Mary G. Colvin, Mika McAnally, Emily Tucker, Li Zhang, Aaron S. Kiess, Thu T. N. Dinh, Anuraj T. Sukumaran

**Affiliations:** 1Department of Poultry Science, Mississippi State University, Mississippi State, MS 39762, USA; 2Prestage Department of Poultry Science, North Carolina State University, Raleigh, NC 27607, USA; 3Tyson Foods, 2200 W. Don Tyson Parkway, Springdale, AR 72762, USA

**Keywords:** *Salmonella*, *Campylobacter*, peracetic acid, poultry processing, prevalence

## Abstract

**Simple Summary:**

*Salmonella* and *Campylobacter* are two of the most common foodborne pathogens isolated from poultry meat. Over the years, a number of advancements have been made in poultry processing to reduce the prevalence of these pathogens, such as the utilization of peracetic acid in various processing steps. However, despite these efforts, *Salmonella* and *Campylobacter* continue to persist in retail broiler meat products. In an effort to characterize the efficacy of existing peracetic acid antimicrobial interventions in the industry, we collected broiler meat samples from throughout the processing chain and from different commercial poultry processing plants. Our results suggest that antimicrobial spray cabinets demonstrate little efficacy in reducing the prevalence of these pathogens. However, the utilization of peracetic acid in carcass chilling tanks remains the most effective chemical intervention. An increase in prevalence during second processing and MDM production suggests that cross-contamination still plays a pivotal role in broiler meat contamination at the retail level.

**Abstract:**

In poultry processing, *Salmonella* and *Campylobacter* contaminations are major food safety concerns. Peracetic acid (PAA) is an antimicrobial commonly used in commercial poultry processing to reduce pathogen prevalence so as to meet the USDA-FSIS performance standards. The objective of this study was to determine the prevalence of *Salmonella* and *Campylobacter* on broiler meat in various steps of commercial poultry processing in plants that use PAA. Post-pick, pre-chill, post-chill, and drumstick chicken samples were collected from three processing plants and mechanically deboned meat (MDM) was collected from two of the three plants. Each plant was sampled thrice, and 10 samples were collected from each processing step during each visit. Among the 420 samples, 79 were contaminated with *Salmonella* and 155 were contaminated with *Campylobacter*. *Salmonella* and *Campylobacter* contamination on the post-pick samples averaged 32.2%. Significant reductions in *Salmonella* and *Campylobacter* were observed in pre-chill to post-chill samples, where the prevalence was reduced from 34% and 64.4% to nondetectable limits and 1.1%, respectively (*p* < 0.001). *Salmonella* and *Campylobacter* remained undetectable on the drumstick samples in all three processing plants. However, the prevalence of *Salmonella* and *Campylobacter* on MDM was similar to the post-pick prevalence, which suggests substantial cross-contamination from post-chill to MDM.

## 1. Introduction

Broiler production alone accounts for 70% of the annual revenue of the poultry industry in the United States [1]. In 2019, 59 billion pounds of broiler meat was processed according to USDA statistical data [1]. A vast number of foodborne infections are related to contaminated poultry meat, with estimates suggesting that poultry is responsible for 25% of outbreaks, illnesses, and hospitalizations [2,3]. Based on these estimations, the consumption of contaminated poultry meat in the United States results in an average of 1.5 million foodborne infections every year [4,5].

One of the major foodborne pathogens commonly isolated in poultry meat is *Salmonella* [3,6]. According to the CDC, *Salmonella* is responsible for an estimated 1.35 million infections and 26,500 hospitalizations in the United States every year [7]. Although there are over 2500 serotypes of *Salmonella*, as listed by the World Health Organization, less than 100 are responsible for the majority of foodborne infections. Some of the most common serotypes historically associated with poultry include *S.* Typhimurium, *S.* Enteritidis, and *S.* Heidelberg [3,8]. However, in recent years, less common serotypes, such as *S.* Reading, *S.* Schwarzengrund, and *S.* Kentucky, have become more prevalent in broiler meat [9,10]. *Salmonella* is found in many phases of broiler production, and it is especially persistent within poultry processing. *Salmonella* is most often introduced into processing plants by live birds and is extremely difficult to irradicate due to cross-contamination, *Salmonella* stress tolerance, residual organic matter, and resistance to sanitation practices [11]. Previous studies have found that cross-contamination is most likely to occur through the exposure of clean broiler meat to equipment that has come in contact with previously contaminated birds [12,13]. Based on previous findings, *Salmonella* prevalence on broiler carcasses in the first processing steps, such as shackling, post-pick, evisceration, and pre-chill, can vary between 30–70% [12,14,15]. However, the prevalence tends to decrease substantially as broiler meat passes through various hurdles.

Another pathogen that continues to be a prominent food safety concern is *Campylobacter*. Ingesting *Campylobacter* often leads to infection, and the USDA reported that doses as low as 500 cells can cause the illness [16]. There are actually more instances of Campylobacteriosis than Salmonellosis each year, with the number of estimated *Campylobacter* infections averaging 1.5 million in the United States [17]. Campylobacteriosis commonly causes fever, abdominal cramps, and bloody diarrhea and may result in hospitalization. In the United States, contaminated broiler meat is responsible for up to 30% of foodborne Campylobacteriosis [18].

There are three species of *Campylobacter* that are commonly associated with poultry related infections: *C*. *jejuni*, *C*. *coli*, and *C*. *lari* [19]. *Campylobacter* requires microaerophilic conditions and elevated temperatures to survive, which is why it flourishes in the G.I. tract of poultry; the gastrointestinal tract of poultry typically contains low levels of oxygen and a higher core temperature of 42 °C [20]. Studies have found a high degree of *Campylobacter* prevalence on broiler meat, with contamination on retail carcasses and chicken breasts ranging from as low as 24% to upwards of 49% [21,22,23,24].

Processing is the last phase of production before products are distributed to retailers. One of the highest priorities for processors is the safety of consumers, i.e., ensuring that the final product is safe and suitable for consumption. In an effort to maintain this standard, it is imperative that antimicrobial interventions are effective in inhibiting the spread and growth of bacteria on broiler meat. Different interventions are utilized in broiler processing to meet the Food Safety Inspection Services performance standards. Antimicrobial compounds are commonly employed at various stages of processing, such as scalding, evisceration, inside-outside bird wash, chilling, and post-chill dip tanks, to meet these safety standards [25]. The antimicrobial used most frequently today is peracetic acid (PAA). With a maximum permissible limit of 2000 ppm per USDA standard, PAA can be applied in spray, dip, or immersion-chilling applications [26].

Although numerous studies have reported the efficacy of PAA against *Salmonella* and *Campylobacter*, there is a lack of recent studies evaluating the efficacy of PAA in commercial settings. By sampling from multiple processing plants, the efficacy of PAA interventions against various microbes can be more accurately measured. There is also a lack of comprehensive studies examining the prevalence of *Salmonella* and *Campylobacter* in processing plants. Much of the existing literature has only observed the microbial load on the end products. However, it may be beneficial to determine the prevalence of *Salmonella* and *Campylobacter* throughout the processing stages in order to identify trends in microbial persistence and survivability, as well as the efficacy of current intervention hurdles. With growing support for “No Antibiotic Ever” farms and the consequential lack of preharvest control, as well as perpetual changes in the industry standards for antimicrobial applications and concentrations, the need for prevalence studies is ever more apparent. 

Therefore, the objectives of this study were (1) to determine the prevalence of *Salmonella* and *Campylobacter* at five stages of poultry processing in commercial processing plants that use different concentrations of PAA, (2) to identify the predominant species of *Campylobacter* in processing plants, and (3) to identify prevalent serotypes of *Salmonella* in commercial broiler processing plants.

## 2. Materials and Methods

### 2.1. Experimental Design

Three commercial processing plants in Mississippi and Alabama that use PAA were selected for this study. Each plant was visited for sampling three times (replications). At each plant, 10 broiler meat samples were collected from each of the 5 processing stages (per replication): post-pick, pre-chill, post-chill, mechanically deboned meat (MDM), and drumsticks. However, 1 of the 3 processing plants did not produce MDM. Thus, MDM data were analyzed independently from the other processing steps. In total, 420 samples were collected. In all 3 processing plants, PAA was the primary antimicrobial intervention. All 3 plants applied PAA through pre-chilling with online-reprocess cabinets (OLR cabinets) at the concentrations ranging from 138–187 ppm. Although at different concentrations, PAA was used in pre-chiller, drag chiller, and finishing chiller tanks for the carcasses, and dip tanks were used for the drumsticks. The greatest difference in the PAA concentration was observed between the plants in the finishing chiller. Plant 1 utilized an average concentration of 767 ppm, plant 2 used an average of 412 ppm, and plant 3 used an average of 705 ppm of PAA in the finishing chiller. Plant 1 was the only plant that employed New York (NY) rinse cabinets post-pick, with the average concentration being 183 ppm. There were no PAA interventions applied to the MDM samples at any of the plants. At each sampling, PAA concentrations were recorded for each sampling step and listed as previously described [27].

### 2.2. Sample Collection

Broiler carcasses from post-pick, pre-chill, and post-chill locations were rinsed with 400 mL of buffered peptone water (BPW) (3M, Saint Paul, MN, USA), as per the USDA isolation guidelines, including MLG 4.10 in 15 in × 20 in 3M sterile bird rinse bags [28]. The rinsate collected from the broiler carcasses was then poured back into the 3M BPW bottles and stored on ice while the drumsticks and MDM samples were collected. Drumsticks were transferred to sterile 750 mL Whirl-Pak bags and MDM was collected in 15 in × 20 in sterile carcass sampling bags. The carcass rinsate, drumsticks, and MDM samples were stored on ice for no more than 3 h during transit back to the Mississippi State Poultry Science Department BSL-2 laboratory. Upon arrival, the 10 drumsticks were rinsed with 225 mL of BPW for 1 min. Twenty-five-gram samples of MDM were each weighed in a new sterile weigh boat, transferred to 750 mL Whirl-Pak^®^ bags (Nasco Sampling/Whirl-Pak^®^, Madison, WI, USA), and homogenized in 225 mL of BPW for 1 min. The BPW rinsate from each sample was used for the isolation of *Salmonella* and *Campylobacter* [28].

### 2.3. Salmonella Isolation

For each sample, 40 mL of BPW rinsate was incubated aerobically at 35 °C for 24 h. After 24 h, 0.5 mL of the rinsate was transferred to 9.5 mL of selective tetrathionate (TT) broth (Becton, Dickinson and Company/Difco^TM^, Sparks, MD, USA) and incubated at 42 °C for 24 h. Following incubation, a loopful of the solution was streaked onto xylose lysine tergitol 4 (XLT4) agar plates (Becton, Dickinson and Company/Difco^TM^, Sparks, MD, USA) and incubated aerobically at 35 °C for 24 h. Positive isolates were identified as black colonies and re-cultured in brain heart infusion broth (BHI). Bacterial cultures were stored at −80 °C in glycerol (1.6 mL culture and 400 µL 80% glycerol).

### 2.4. Salmonella DNA Extraction and PCR Confirmation

The isolates were re-cultured by inoculating 10mL of BHI broth with frozen stock cultures. The cultures were incubated at 37 °C for 24 h and then pelleted by centrifuging in an Eppendorf centrifuge 5810 R (Eppendorf, Hamburg, Germany) at 4000 rpm for 15 min. The extraction of *Salmonella* DNA was performed using a Thermo Fisher Genejet genomic DNA extraction kit K0721 (Thermo Fisher Scientific, Inc., Waltham, MA, USA), and the purity of each sample was determined using a NanoDrop One (Thermo Fisher Scientific, Madison, WI, USA). Traditional PCR was used to amplify the targeted *invA* gene using a PCR Thermo Cycler (Model 5435 Mastercycler epgradient S, Eppendorf, Hamburg, Germany). *Salmonella*-specific primers, S139 and S141, which were used as described by Rahn et al. (1992), have the following nucleotide sequences: 5′-GTG AAA TTA TCG CCA CGT TCG GGC AA-3′ and 5′-TCA TCG CAC CGT CAA AGG AAC C-3′ [28]. The PCR reaction mixture consisted of 1 µL of DNA and 9 µL of master mix containing 5 µL of GoTaq Green (Promega, Madison, WI, USA), 0.25 µL of each forward and reverse primer, and 3.5 µL of molecular-grade water. The cycle conditions for PCR were as follows: an initial denaturation step at 94 °C for 3 min, followed by 35 cycles of denaturation at 94 °C for 1 min, annealing at 53 °C for 2 min, and primer extension at 72 °C for 3 min. A final incubation cycle was set at 72 °C for 7 min based on previous findings [29]. The *invA* gene from the ATCC *S.* Typhimurium strain 14,028, was used as a positive control. The PCR master mix alone served as a negative control.

### 2.5. Electrophoresis of the PCR Products

To confirm the amplification of the target gene, PCR products (284 bp DNA fragments) were electrophoresed on a 2% agarose gel containing SYBR™ Safe DNA gel stain (Invitrogen™, Carlsbad, CA, USA) in 1X Tris-Acetate-EDTA (TAE) buffer. The products were visualized under UV light using the Kodak Gel Logic 200 Imaging System (Eastman Kodak Co., Rochester, NY, USA). A current of 80 V was applied to each gel. The PCR product (3 µL) was loaded into each well of the gel and a 100 bp DNA ladder was used as a marker for the PCR products.

### 2.6. Serotyping

Isolates confirmed to be *Salmonella* by PCR were sent to the National Veterinary Services laboratory, Ames, Iowa for serotyping. Tryptic soy agar (TSA) slants were prepared by pouring 10 mL of agar into 15 mL screw-top tubes, and each isolate was streaked onto the surface of a slant and incubated at 37 °C for 18–20 h. The agar slants were secured in a cooler with a 2 lb block of dry ice and mailed within 2 weeks of the slant preparation.

### 2.7. Campylobacter Isolation

For each sample, 20 mL of BPW rinsate was enriched in 20 mL Bolton’s broth (2XBFBEB) (Oxoid Ltd., Hasingstoke, UK) by incubating at 42 °C for 48 h in microaerophilic conditions. The enriched rinsate was streaked onto Campy-Cefex agar (Oxoid Ltd., Hasingstoke, Hants, UK) plates and incubated at 42 °C for 48 h in microaerophilic conditions. Positive isolates were identified as 0.5–2 mm grey colonies and re-cultured in BHI. Bacteria cultures were stored at −80 °C in glycerol (1.5 mL culture and 400 µL 80% glycerol).

### 2.8. Campylobacter DNA Extraction and PCR

The samples were prepared by culturing the presumed positive isolates in 10 mL of brain heart infusion (BHI) broth and incubating at 42 °C for 48 h in microaerophilic conditions. After 48 h, cultures were centrifuged in an Eppendorf centrifuge 5810 R (Eppendorf, Hamburg, Germany) at 4000 rpm for 15 min. The supernatant was pipetted out, and the bacterial pellet was used for the DNA extraction. The extraction of *Campylobacter* DNA was performed using a Thermo Fisher Genejet genomic DNA extraction kit K0721 (Thermo Fisher Scientific Baltics UAB, Vilnius, Lithuania), and the purity of each sample was confirmed using a NanoDrop One (Thermo Fisher Scientific, Madison, WI, USA). Traditional PCR was used to test the isolates for the 3 strains of *Campylobacter*, *C*. *jejuni*, *C*. *coli*, and *C*. *lari*. Three pairs of primers were selected to identify the genes *hipO* from *C*. *jejuni*, *glyA* from *C*. *coli*, and *glyA* from *C*. *lari*, as previously described [29]. The primer sequences used in each PCR reaction are outlined in Table 1. For each PCR reaction, 1µL of DNA and 9µL of master mix, containing 5.0 µL of GoTaq Green, 0.25 µL of forward and reverse primers, and 3.5 µL of molecular-grade water were aliquoted into the wells of a 96-well plate (Applied Biosystems, Life Technologies Holdings Pte Ltd., Singapore). The cycle conditions for PCR were as follows: an initial denaturation step at 95 °C for 2 min followed by 35 cycles of denaturation at 95 °C for 30 s, annealing at 58 °C, 55 °C, and 55 °C for *C*. jejuni, *C*. coli, and *C*. lari, respectively, for 30 s, a primer extension at 72 °C for 30 s, and a final extension at 72 °C for 5 min [30].

### 2.9. Electrophoresis of the PCR Products

To confirm the amplification of the DNA from the *Campylobacter-*species-specific PCR, the products were analyzed using electrophoresis on a 1.5% agarose gel containing SYBR™ Safe DNA gel stain (Invitrogen™, Carlsbad, CA, USA) and visualized under UV light. A current of 80 V was applied to each gel. Two µL of PCR product was loaded into each well of the gel, and a 100 bp DNA ladder (Thermo Scientific, Vilnius, Lithuania) was used as a marker for the PCR products.

### 2.10. Statistical Analysis

A completely randomized design with a 3 (processing plants) × 4 (processing steps) factorial arrangement was used for this study. One of the three processing plants did not produce MDM. Therefore, the microbial prevalence on MDM was analyzed separately using Student’s *t*-test to determine the differences between processing plants. Data were analyzed using the GLIMMIX procedure of SAS version 9.4 (SAS Institute Inc., Cary, NC, USA). Statistical differences were determined by a protected *t*-test using the LSMEANS procedure and statistical significance was established at a *p* value of ≤0.05.

## 3. Results

### 3.1. Salmonella Prevalence

Of the 420 samples collected, 79 (18.8%) contained *Salmonella*. There were no differences in *Salmonella* prevalence between the processing plants (*p* = 0.633). However, as seen in Figure 1, the *Salmonella* prevalence was affected by the processing step (*p* = 0.002). No significant differences were observed between the post-pick and pre-chill steps (*p* = 0.832). The prevalence of *Salmonella* post-pick and pre-chill averaged 32% and 34%, respectively. The *Salmonella* prevalence was reduced by 34% to non-detectable levels in the post-chill carcasses and remained undetectable in the drumsticks. As stated earlier, MDM was analyzed independently because it was only produced in plant 1 and plant 3. As seen in Figure 2, the rates of prevalence of *Salmonella* in MDM at plant 1 and plant 3 were similar and averaged 33% and 30%, respectively (*p* = 0.909). A total of 12 *Salmonella* samples were selected for serotyping based on the purity and morphology of the individual colonies that were collected. Two serotypes were identified from 12 samples submitted to the National Veterinary Services Laboratories: *S*. Kentucky was present in three samples and *S*. Schwarzengrund was present in nine samples. *S.* Kentucky was only identified on MDM samples, whereas *S.* Schwarzengrund was found on post-pick, pre-chill, and MDM samples.

### 3.2. Campylobacter Prevalence

Of the 420 samples collected, 155 (36.9%) were contaminated with *Campylobacter*. Among the 155 isolates confirmed by PCR, 97 (62.3%) and 58 (37.4%) were *C*. *jejuni* and *C*. *coli*, respectively. No *C*. lari was detected. No differences were observed in *Campylobacter* prevalence between the processing plants (*p* = 0.201). However, as seen in Figure 3, *Campylobacter* prevalence was affected by the processing step (*p* < 0.001). No significant differences were observed between the post-pick and pre-chill carcasses (*p* = 0.615). The rates of prevalence of *Campylobacter* in the post-pick and pre-chill samples were 71.1% and 64.4%, respectively. However, *Campylobacter* prevalence was significantly reduced from pre-chill to post-chill by 63.3%, on average (*p* < 0.001). Although not considered statistically significant, an average of 4.4% of the drumstick samples were contaminated with *Campylobacter*. As mentioned previously, MDM was analyzed independently from the other steps because only plant 1 and plant 3 produced MDM. As seen in Figure 4, the prevalence of *Campylobacter* on MDM in plant 3 was 60%, as compared to 33.3% in plant 1 (*p* = 0.304).

## 4. Discussion

### 4.1. Salmonella Contamination

*Salmonella* contamination on the broiler carcasses in this study was similar to other findings. In this study, the initial *Salmonella* contamination averaged 32%, whereas previously, it was found to be at an average of 40% [31]. Additionally, this is congruent with previous research, which found *Salmonella* contamination on first processing equipment to be at an average of 40% [9]. Based on the results, all three processing plants in this study demonstrated a reduction in *Salmonella* overall. The reduction in *Salmonella* prevalence on the broiler carcasses in this study echoes findings reported by Nagel et al., 2013, Bailey et al., 2020., and Kumar et al., 2020, with most recent reports finding that the prevalence of *Salmonella* on carcasses was reduced from 38% to 5% [32,33,34]. By comparison, we observed that *Salmonella* was reduced from 34% to non-detectable levels on carcasses. A surprising finding of this study was the low percentage of isolates confirmed to be *Salmonella*. Although black colonies were selected and re-cultured for confirmation, less than half were confirmed by traditional PCR. Previous studies have discussed the possibility of other Gram-negative bacterial growth on XLT4 media [35]. Even though the likelihood of this is low, the potential for unwarranted growth may be exacerbated by the degree of contamination from the field samples. Likewise, similar reports can be found that warn against using visual media confirmation [36]. Therefore, it is necessary to reiterate the importance of using molecular confirmation for species determination, as it is significantly more reliable.

### 4.2. Campylobacter Contamination

Based on the results, all three processing plants in this study demonstrated a reduction in the prevalence of *Campylobacter* during processing. The initial *Campylobacter* contamination on carcasses was similar to previous findings [32]. In this study, carcass contamination at post-pick sampling averaged 71%, whereas Nagel et al. (2013) found even higher contamination levels in various plants, averaging above 90%. This is similar to findings where *Campylobacter* prevalence at post-pick was found to be 100% among three flocks of birds [37]. Similar to this study, McCrea et al., 2006, found no major differences between the post-pick and pre-chill rates of *Campylobacter* prevalence. Carcass contamination was reduced from 100% to 99%, which is less of a reduction than that found in this study [37]. These results suggest the application methods of PAA are not effective in reducing *Campylobacter* prevalence during the early stages of processing. Despite the use of additional NY rinse cabinets in plant 1 and the use of OLR cabinets on pre-chill samples, the prevalence of *Campylobacter* was not significantly different (*p* = 0.585). However, significant reductions in *Campylobacter* during post-chill sampling in this study agree with the results reported by Nagel et al., 2013, and Wideman et al., 2016, who found that PAA was successful in reducing *Campylobacter* on carcasses in chilling tanks by 0.8 log CFU/mL and 2.03 log CFU/mL, respectively [5,32]. By comparison, in this study, *Campylobacter* prevalence was reduced from 64.4% at pre-chill to 1.1% after submersion in the carcass chilling tanks. An interesting finding, observing *Campylobacter* prevalence reports, is that the methodology used to determine *Campylobacter* prevalence during broiler processing varies significantly between publications [38]. Out of 111 articles published between 1983–2010, only 32 examine the prevalence or concentration of *Campylobacter* at different stages of processing [38]. Although the general trend in the data suggests a high initial carcass contamination, contamination decreases after chilling [38].

### 4.3. MDM Contamination

Although the rates of prevalence of *Salmonella* and *Campylobacter* were significantly lower in the post-chill and drumstick samples, the prevalence of these pathogens in the MDM samples was as high as it was in the post-pick samples. Currently, mechanically deboned meat is not federally inspected for *Salmonella* prevalence. One area of concern regarding this practice is the use of comminuted chicken for MDM. As of now, the maximum acceptable % of product tested positive for *Salmonella* on comminuted chicken is 25% [39]. Comminuted chicken utilized in MDM products would risk failing to meet the performance standards if there was a failure in the heat treatment applications. Based on these results and previous findings, data would suggest that high rates of cross-contamination occur while mechanically deboning meat due to increased contact with additional surfaces, processing equipment, and broiler meat [40,41,42]. Other previous research suggests that the higher contamination of mechanically deboned meat may be attributed to pH differences between meat used for retail cuts and meat that is mechanically separated, which is closely attached to the skeleton [43]. Previous research suggests the higher pH in mechanically separated meat reduces the effectiveness of interventions used to destroy bacterial cell membranes [43]. Due to the levels of contamination in MDM found in this study, it may be beneficial to begin investigating potential interventions during this stage of processing. Not only could this minimize food safety risks for consumers and affect the product shelf life, but a proactive approach may also alleviate processing restrictions, time restrictions, and profit losses should MDM inspection be required in the future. Although a standardized treatment for MDM has not been proposed for commercial use, some findings have displayed potential. A study in 2016 investigated dip applications of peracetic acid and cetylpyridinium chloride at 0.1% and 0.5%, respectively, to carcass frames immediately prior to mechanical separation. It was found that dip applications of CPC at 60 s completely reduced *Salmonella* from 33% to nondetectable levels [44]. Comparatively, PAA at 90 s reduced *Salmonella* from 93% to 50% [44].

### 4.4. Peracetic Acid Interventions

Surprisingly, there were no major differences in prevalence between the post-pick and pre-chill steps despite the use of NY rinse cabinets and OLR spray cabinets. Previous findings have discussed a variety of results for samples between post-pick and pre-chill sampling, with prevalence rates ranging from 20–40%, with no major reductions [38]. In some instances, it was observed that the prevalence of *Salmonella* and *Campylobacter* actually increased on pre-chill samples from 0.79 to 1.4 log CFU/mL [5]. A similar trend was observed for *Salmonella* prevalence in this study, and the differences in *Campylobacter* prevalence were not statistically significant. Based on the findings of this study and the wide range of results reported in previous research, antimicrobial spray cabinets during post-pick and pre-chill sampling are less effective. Articles collected over the last 30 years would suggest, at the very least, that the effect of antimicrobial spray cabinets commonly used during these steps is inconsistent [45]. One of the primary functions of spray cabinets prior to the use of chilling tanks is to remove visible contamination from the carcasses [46]. Spray cabinets should, in turn, reduce microbial contamination as well. One of the potential shortcomings of this study is that bacterial concentrations were not recorded. It is possible that, while there were no differences between the post-pick and pre-chill prevalence of *Salmonella* and *Campylobacter*, there may have been differences in the concentrations. Previous research has suggested that OLR spray cabinets reduce *Salmonella* prevalence from 34% to 28% [46]. Although the observed differences in prevalence were not significant, the quantification of the bacterial counts demonstrated an average reduction of 0.5 log CFU/mL, which was considered significant [46]. One potential explanation for this discrepancy has been attributed to the significant cross-contamination that occurs during the initial processing steps [47,48,49]. The mechanism of *Salmonella* and *Campylobacter* persistence on carcass skin is not fully understood, as there is limited data characterizing the locations on carcasses that harbor naturally occurring *Salmonella* and *Campylobacter* [48]. It was apparent, in this study, that the use of carcass chilling tanks with PAA was most effective at reducing the prevalence of both *Salmonella* and *Campylobacter*. As mentioned previously, these results are similar to other findings of studies which investigated the efficacy of PAA in carcass chillers [50,51]. This may be due to the conditions in chiller tanks, which are ideal for PAA activation. As explained by Kataria et al., 2020, PAA inhibits protein synthesis by altering the cell membrane permeability [50]. Chilling tanks allow carcasses to be exposed for prolonged periods of time, thus improving the exposure of pathogenic bacteria to the antimicrobial. Peracetic acid also has the added benefit of maintaining its effectiveness in the presence of organic matter. These factors may contribute towards greater reductions in bacterial prevalence, as well as the overall microbial load, in carcass chiller tanks. There was some variation in the concentration of PAA used between plants [27]. However, despite the organizational differences between processing plants and variation in PAA concentrations, no plant effect or plant-step interaction were observed. Therefore, given that neither the processing plant nor concentration had any effect on the prevalence, we recommend using the lowest effective concentration in the chilling tanks. Lowering the concentration in these steps may be more economical and may better for preserving the sensory characteristics of broiler meat [32].

## 5. Conclusions

The results of this study indicate that PAA was effective in reducing the prevalence of *Salmonella* and *Campylobacter*. However, only the chilling tanks significantly reduced both *Salmonella* and *Campylobacter* compared to the other steps. No significant differences were observed between processing plants. Therefore, we would recommend using the lowest effective concentration of PAA in finishing chillers in order to reduce costs and preserve the sensory characteristics of meat products. Significant differences between post-chill and MDM contamination levels suggest high levels of cross-contamination occur during this step, and additional measures aiming to reduce *Salmonella* and *Campylobacter* in MDM may be worth investigating. Lastly, molecular confirmation should be used for species determination, as it is more reliable than visual verification using selective media.

## Figures and Tables

**Figure 1 animals-12-02460-f001:**
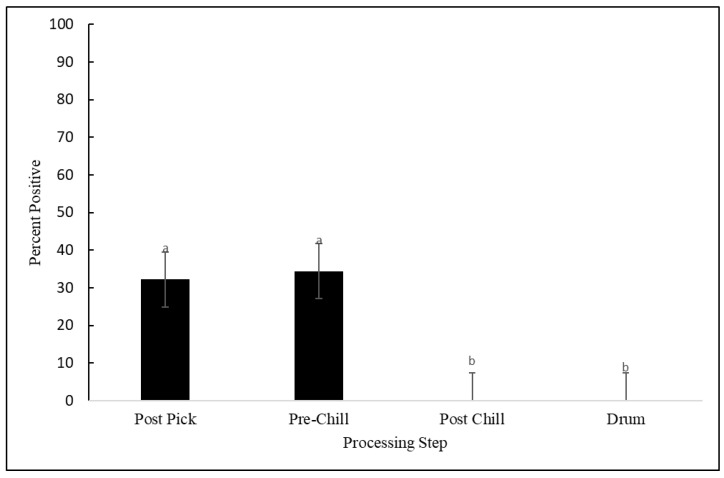
Prevalence of *Salmonella* as a percentage, detected on the samples collected at the post-pick, pre-chill, and post-chill stages, and drumsticks. (*p* = 0.002). The error bars represent pooled standard errors of the means. Means with different superscripts differ statistically.

**Figure 2 animals-12-02460-f002:**
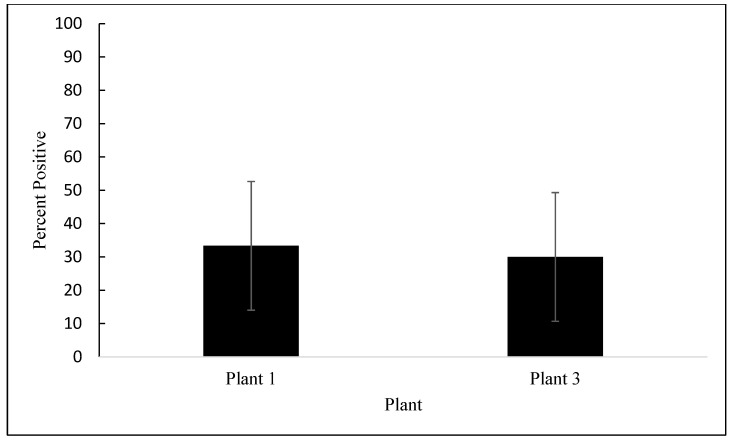
Prevalence of *Salmonella* as a percentage, detected on MDM samples in processing plant 1 and plant 3. (*p* = 0.9087). The error bars represent pooled standard errors of the means.

**Figure 3 animals-12-02460-f003:**
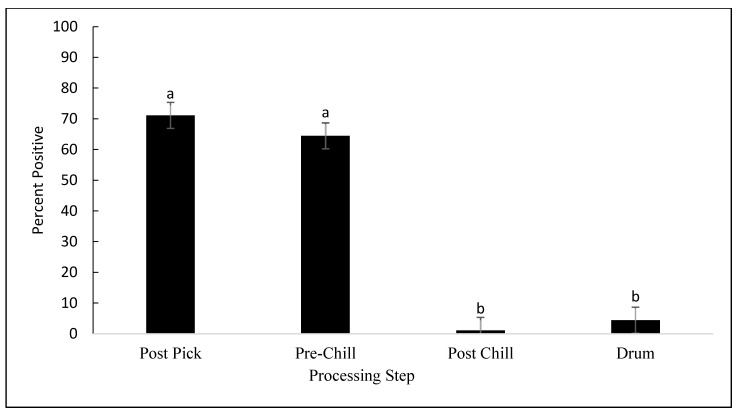
Prevalence of *Campylobacter* as a percentage, detected on samples collected at post-pick, pre-chill, and post-chill, and drumsticks (*p* < 0.001). The error bars represent pooled standard errors of the means. Means with different superscripts differ statistically.

**Figure 4 animals-12-02460-f004:**
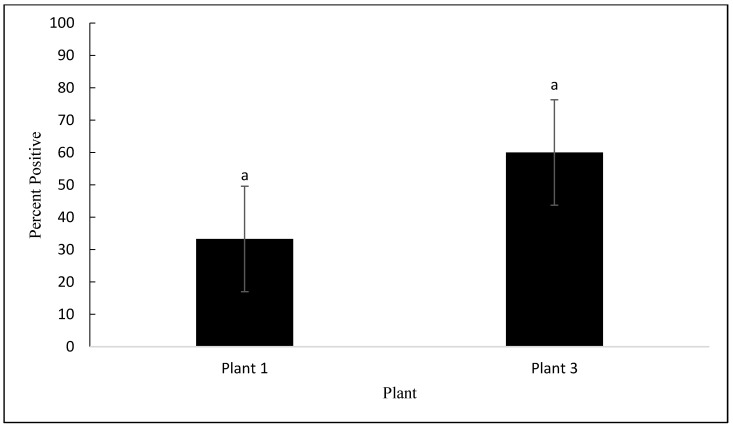
Prevalence of *Campylobacter* as a percentage, detected on the MDM samples in processing plant 1 and plant 3 (*p* = 0.304). The error bars represent pooled standard errors of the means. Means with different superscripts differ statistically.

**Table 1 animals-12-02460-t001:** Primer sequences used in the PCR assays and the expected sizes of the products.

Primer	Size (in bp)	Sequence (5′-3′)	GenBank Accession No.	Target Gene	Gene Location (bp)
CJF	323	ACTTCTTTATTGCTTGCTGC	Z36940	*C. jejuni hipO*	1662–1681
CJR		GCCACAACAAGTAAAGAAGC			1984–1965
CCF	126	GTAAAACCAAAGCTTATCGTG	AF136494	*C. coli* glyA	337–357
CCR		TCCAGCAATGTGTGCAATG			462–444
CLF	251	TAGAGAGATAGCAAAAGAGA	AF136495	*C. lari* glyA	318–337
CLR		TACACATAATAATCCCACCC			568–549

## Data Availability

The data presented in this study are available on request from the corresponding author.

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
