# Peer review of "The Prevalence of Salmonella and Campylobacter on Broiler Meat at Different Stages of Commercial Poultry Processing"

_animals, 2022, doi:10.3390/ani12182460_

Round 1
Reviewer 1 Report
General Comments: The manuscript is well written and although the work is not particularly novel, it does provide some useful information.
Specific Comments:
L51-54 This combination of sentences seems to suggest that these are the serotypes most often causing illness. This is not true for Kentucky.
L84-86 Antimicrobials are not often used during defeathering. They are more often used in the IOBW, OLR, and dip tanks in addition to the other locations mentioned.
L99 Lack of preharvest control is not a consequence of NAE. Growth promoting antibiotics were not effective against Salmonella and Campylobacter when applied.
L126 Why was a neutralizing BPW not used? PAA has been clearly demonstrated to lead to further reductions within rinsates after sampling.
Both Salmonella and Campylobacter isolation – why were FSIS MLG methods not followed? Only using TT and XLT4 for Salmonella can skew they serotypes which are identified. Why was only 20 mL used for Campylobacter enrichment?
L253 The abstract states that Campylobacter was below detectible levels.
L254 roughly? Is 4.4% not the calculated mean?
In each separate figure, the standard errors appear identical. Is this accurate? What are the standard errors? They do not appear to be mentioned in the text.
L282-283 Provide a reference. Using other tests such as agglutination, TSI/LIA slants are also useful and are mentioned in the FSIS MLG, which you did not follow.
L336 You do not know if it had an effect. You only know that it was not different than post-pick. The OLR could have reduced prevalence that had increased during evisceration.
There does not appear to be any mention of why PAA in chilling was most effective.
L379-380 Again the authors appear to have overlooked about the other techniques for confirming Salmonella.
Author Response
Reviewer 1
Comments and Suggestions for Authors
General Comments: The manuscript is well written and although the work is not particularly novel, it does provide some useful information.
Specific Comments:
L51-54 This combination of sentences seems to suggest that these are the serotypes most often causing illness. This is not true for Kentucky.
We thank the reviewer for this comment. The intent of this section was to iterate that in general, a small number of Salmonella serotypes are responsible for human infection. While historically there are several common serotypes, there is an increased incidence of less commonly isolated serotypes. In an effort to better convey this, changes were made to lines 53-54 to highlight this key point. The inclusion of S. Kentucky as a major poultry contaminant in this statement is supported by USDA-FSIS report seen here:
U.S. Department of Agriculture, Food Safety and Inspection Service. 2016. Serotypes profile of Salmonella isolates from meat and poultry products, January 1998 through December 2014. Available at: https://www.fsis.usda.gov/wps/portal/fsis/topics/datacollection-and-reports/microbiology/annual-serotyping-reports.
L84-86 Antimicrobials are not often used during defeathering. They are more often used in the IOBW, OLR, and dip tanks in addition to the other locations mentioned.
Inclusion of “defeathering” in line 85 was removed and the sentence was modified.
L99 Lack of preharvest control is not a consequence of NAE. Growth promoting antibiotics were not effective against Salmonella and Campylobacter when applied.
The purpose of line 99 is to highlight the need for prevalence studies as industry standards have changed over the years. Since interventions like growth promoting antibiotics are not allowed in NAE farms, the phrase “lack of preharvest control” was used, as it is used in several studies investigating NAE farms.
Moreover, several published studies have reported the impact of in-feed antibiotic supplementation in reducing Salmonella colonization in poultry. Please see a review paper for example.
Bailey, J. S. (1988). Integrated colonization control of Salmonella in poultry. Poultry Science, 67(6), 928-932.
L126 Why was a neutralizing BPW not used? PAA has been clearly demonstrated to lead to further reductions within rinsates after sampling.
This is a great comment. While nBPW is used by the USDA, it is not specifically listed under MLG guidelines. We agree with the reviewer that residual PAA can impact the counts, however, in our study, the carcasses were drained properly before rinsing with BPW. Moreover, we believe the sufficiently high volume of BPW used will nullify the residual impact of PAA. Also, there are several published studies that have used BPW to enumerate bacteria on chicken samples after treatment with PAA.
Both Salmonella and Campylobacter isolation – why were FSIS MLG methods not followed? Only using TT and XLT4 for Salmonella can skew they serotypes which are identified. Why was only 20 mL used for Campylobacter enrichment?
Based on previous literature it was decided that only TT broth and XLT4 media would be used, as numerous studies utilize these two selective medias or some variation of MLG guidelines:
Velasquez, C. G., Macklin, K. S., Kumar, S., Bailey, M., Ebner, P. E., Oliver, H. F., ... & Singh, M. (2018). Prevalence and antimicrobial resistance patterns of Salmonella isolated from poultry farms in southeastern United States. Poultry science, 97(6), 2144-2152.
Wideman, N., Bailey, M., Bilgili, S. F., Thippareddi, H., Wang, L., Bratcher, C., ... & Singh, M. (2016). Evaluating best practices for Campylobacter and Salmonella reduction in poultry processing plants. Poultry science, 95(2), 306-315.
Guran, H. S., Mann, D., & Alali, W. Q. (2017). Salmonella prevalence associated with chicken parts with and without skin from retail establishments in Atlanta metropolitan area, Georgia. Food Control, 73, 462-467.
As noted by the reviewer, 20mL of BPW rinsate was enriched with 20mL Bolton’s broth. This mixture was used in order to place all of the contents in a 50mL tube. The ratio from MLG guidelines was kept the same.
L253 The abstract states that Campylobacter was below detectible levels.
We thank the reviewer for this comment. This error was corrected in lines 32-33 where the specific percentage (1.1% prevalence) was listed for Campylobacter
L254 roughly? Is 4.4% not the calculated mean?
The word “roughly” was removed as it does not accurately reflect the calculated mean. This was replaced with “an average of 4.4%”. Please see line 267 of the revised manuscript.
In each separate figure, the standard errors appear identical. Is this accurate? What are the standard errors? They do not appear to be mentioned in the text.
Thanks for this comment. We have added the explanation in figure titles. Yes, the error bars were identical for each figure since we used pooled standard error.
L282-283 Provide a reference. Using other tests such as agglutination, TSI/LIA slants are also useful and are mentioned in the FSIS MLG, which you did not follow.
Reference 34 and 35 were included to highlight the discrepancy found in this study. While TSI/LIA slants were not mentioned in this study, molecular confirmation techniques were encouraged as an efficient and effective confirmation technique.
L336 You do not know if it had an effect. You only know that it was not different than post-pick. The OLR could have reduced prevalence that had increased during evisceration.
L336 was altered to express that there were no differences in prevalence observed between these steps despite the use of NY rinse and OLR cabinets. Please see lines 350-351 of the revised manuscript.
There does not appear to be any mention of why PAA in chilling was most effective.
We thank the reviewer for noticing this opportunity for added discussion. A brief explanation to the antimicrobial activity and effectiveness of PAA in chilling was added in lines 372-378.
L379-380 Again the authors appear to have overlooked about the other techniques for confirming Salmonella.
We thank the reviewer for acknowledging this detail. While other traditional techniques such as biochemical tests are useful, molecular confirmation was emphasized as an efficient and effective method that has developed to be a leading standard.
Reviewer 2 Report
It was a pleasure to read this interesting manuscript investigating Salmonella and Campylobacter in broiler meat at different stages of the commercial processing. The science and analysis is sound, and will be beneficial to the scientific literature, as well as to public health.
I have a few very minor comments which I have detailed below
Keywords- Salmonella and Campylobacter need italicising
Line 57- perhaps mentioning other factors in here maybe useful, just a few examples?
Line 99- as processing here is different to in the US, do ‘no antibiotic ever’ chickens still get the PAA treatments?
Line 110- were these different animals? I am guessing so, but maybe worth saying
Line 127- not completely sure that makes total sense. Please reword
Line 145- maybe bacterial rather than bacteria culture may sound better
Line 146 and 185- is this 80% glycerol? It doesn’t seem it based on the calculations
Line 164- could you add in the product size here, and mention the positive controls and what they were?
Line 166-172- not sure that this is needed or in so much detail but that’s up to the authors
Line 207- you have a bit of the title missing here but that could be during editing
Line 207-213- again not sure that this is needed or in so much detail but that’s up to the authors
Line 266- contamination doesn’t need a capital letter
Line 277- any ideas what the other colonies were?
Line 2780 Gram needs capitalising
Author Response
Reviewer 2
Comments and Suggestions for Authors
It was a pleasure to read this interesting manuscript investigating Salmonella and Campylobacter in broiler meat at different stages of the commercial processing. The science and analysis is sound, and will be beneficial to the scientific literature, as well as to public health.
I have a few very minor comments which I have detailed below
Keywords- Salmonella and Campylobacter need italicizing
We thank the reviewer for this comment. Both Salmonella and Campylobacter were italicized in line 37.
Line 57- perhaps mentioning other factors in here maybe useful, just a few examples?
The phrase “other factors” was removed in line 57 and replaced with “Salmonella stress tolerance, residual organic matter, and resistance to sanitation practices”.
Line 99- as processing here is different to in the US, do ‘no antibiotic ever’ chickens still get the PAA treatments?
‘no antibiotic ever’ claims are described by the USDA as being raised without antibiotics from birth to harvest. The USDA does not explicitly refer to post-harvest control methods in labeling standards for ‘no antibiotic ever’. As each plant is different, some may not use PAA as the primary antimicrobial. However, as PAA is not an antibiotic, it would qualify as a post-harvest treatment for ‘no antibiotic ever’ broiler meat under USDA standards.
https://www.fsis.usda.gov/sites/default/files/media_file/2021-07/FSIS-GD-2019-0004-slide.pdf
Line 110- were these different animals? I am guessing so, but maybe worth saying
“broiler meat samples” was added in line 113 to differentiate between individual samples. Each sample was an individual carcass, drumstick, or batch of MDM but this was excluded to prevent confusion, as the methodology already describes the sampling protocols.
Line 127- not completely sure that makes total sense. Please reword
In line 129, the sentence was restructured to “The rinsate collected from broiler carcasses was then poured back into the 3M BPW bottles…”. As the product listed in 127 describes the specific BPW container, line 129 is used to indicate the bottles were used to store the carcass rinsate.
Line 145- maybe bacterial rather than bacteria culture may sound better
Bacteria was converted to “Bacterial” in line 147.
Line 146 and 185- is this 80% glycerol? It doesn’t seem it based on the calculations
We thank the reviewer for this comment. “80%” was moved to within the parenthesis in line 148 and line 187. 400µL of 80% glycerol and 1.6 mL of culture was used.
Line 164- could you add in the product size here, and mention the positive controls and what they were?
The product size was added in line 169 (284bp DNA fragment), and the positive control used was described in line 166 as the invA gene from ATCC strain 14028.
Line 166-172- not sure that this is needed or in so much detail but that’s up to the authors
To improve the repeatability of the experiment, the Electrophoresis conditions were included.
Line 207- you have a bit of the title missing here but that could be during editing
The title was corrected to display Electophoresis
Line 207-213- again not sure that this is needed or in so much detail but that’s up to the authors
The details of the Electrophoresis conditions were included to improve repeatability for future studies.
Line 266- contamination doesn’t need a capital letter
We thank the reviewer for this comment. Contamination was lower cased in line 269.
Line 277- any ideas what the other colonies were?
Several samples initially sent for serotyping were classified as proteus mirabilis. However, the colonies were not further investigated.
Line 278 Gram needs capitalising
We thank the reviewer for this comment. Gram was capitalized in line 281.
Reviewer 3 Report
Application of peracetic acid in food production (e.g.dairy milk production) remains the most effective intervention step. Basic cleaning chemicals are not enough to achieve full sanitization, so chemical disinfectant, such as Peracetic Acid must be used. Chemical treatment with Peracetic Acid is among the most effective methods for control of microbial contamination.
Results from this study indicate that PAA was reduced the prevalence of Salmonella and Campylobacter. This fact is well known, practiced in processing plants. Author concluded that only the chilling tanks significantly reduced both Salmonella and Campylobacter compared to the other steps. Its also well known fact, especially in termophilic Campylobacter. The collection of a 420 samples and detection of Salmonella and Campylobacter does usually not justify a publication in an international journal. The article is superficial, not detailed enough.
Major revision:
The major omission, which probably and unfortunately cannot be remedied, is the lack of serotyping of all 79 Salmonella strains. The second major omission, is lack of in vitro test of PAA inactivity against Salmonella strains or Campylobacter. Moreover, which concentration was most efficient?
Minor revisions:
Line 26: drumstick chicken
Line 125: Why rinsate? Better will be meat samples from broiler or drumstick (like MDM).
Line 136: To isolation Campylobacter authors used BPW? Not Bolton buffer?
Line 142: Following incubation, a loop full of the solution was streaked onto xylose lysine tergitol 4. Why directly to solid agar, not to semi-solid MSRV agar?
Line 164: Positive control and negative consist of?
Line 172: lack of PCR product size
lINE 197: Wang et al., 2002 [citation]
Line 207: Electrophoresis
Author Response
Reviewer 3
Comments and Suggestions for Authors
Application of peracetic acid in food production (e.g.dairy milk production) remains the most effective intervention step. Basic cleaning chemicals are not enough to achieve full sanitization, so chemical disinfectant, such as Peracetic Acid must be used. Chemical treatment with Peracetic Acid is among the most effective methods for control of microbial contamination.
Results from this study indicate that PAA was reduced the prevalence of Salmonella and Campylobacter. This fact is well known, practiced in processing plants. Author concluded that only the chilling tanks significantly reduced both Salmonella and Campylobacter compared to the other steps. Its also well known fact, especially in termophilic Campylobacter. The collection of a 420 samples and detection of Salmonella and Campylobacter does usually not justify a publication in an international journal. The article is superficial, not detailed enough.
We thank the reviewer for expressing his/her overall view of the manuscript. However, we disagree with the comment about insufficient sample size. The sample size of 420 used in this study was appropriate and was based on proper power analysis and previous studies. We agree that several studies have tested and proved the efficacy of PAA against these pathogens, however limited studies have investigated the actual industry prevalence of these pathogens in commercial plants that use PAA. Therefore, this study is unique.
Major revision:
The major omission, which probably and unfortunately cannot be remedied, is the lack of serotyping of all 79 Salmonella strains. The second major omission, is lack of in vitro test of PAA inactivity against Salmonella strains or Campylobacter. Moreover, which concentration was most efficient?
It would have been great if we could have serotyped all the 79 isolates, however, this could not be completed due to several reasons including cost. And, we agree with the reviewer that serotyping all of them would definitely improve the depth of the paper.
Several published studies have already tested the in-vitro efficacy of PAA at different concentrations against both Salmonella and Campylobacter. However, the purpose of our study was to see the efficacy of PAA application in a commercial setting. Because there are several differences between in-vitro testing and real-world applications. Our study is one of the very few studies which have investigated the efficacy of PAA in commercial poultry processing plants at different stages.
Minor revisions:
Line 26: drumstick chicken
Line 27 was altered to clarity “drumstick chicken samples”.
Line 125: Why rinsate? Better will be meat samples from broiler or drumstick (like MDM).
As per USDA MLG 4.10 guidelines which were used at the time, sample rinsate is the recommended
Line 136: To isolation Campylobacter authors used BPW? Not Bolton buffer?
Bolton’s broth was used. Line 183 describes the methodology in which 20mL of BPW rinsate was enriched with 20mL of Bolton’s broth as a modification of USDA MLG guidelines.
Line 142: Following incubation, a loop full of the solution was streaked onto xylose lysine tergitol 4. Why directly to solid agar, not to semi-solid MSRV agar?
This comment is not clear. In our knowledge, there are no such requirements to use a semisolid agar for Salmonella isolation. We followed the USDA MLG 4.10 guidelines with recommended modifications.
Line 164: Positive control and negative consist of?
The details of the positive and negative controls were added in lines 166-168. The positive control was the amplified invA gene from the ATCC S. Typhimurium strain 14028 and the negative control consisted of only the PCR master mix and elution buffer.
Line 172: lack of PCR product size
The PCR product size (284bp) was added in line 170.
lINE 197: Wang et al., 2002 [citation]
The citation was replaced with the citation number.
Line 207: Electrophoresis
The title was fixed to display Electrophoresis.
Round 2
Reviewer 3 Report
Major revision:
Authors comment that: The sample size of 420 used in this study was appropriate and was based on proper power analysis and previous studies.
These 420 samples were taken from only 3 processing plants, thus may be sufficient for statistical analysis, but not to global interpretation.
Authors comment: limited studies have investigated the actual industry prevalence of these pathogens in commercial plants that use PAA. We disagree with the comment. Therefore, this study is not unique. Useing PAA in meat plants is common practice. Australia and New Zealand Food Standards Code allows sanitizers such as peroxyacetic acid (PAA), and sodium hypochlorite for use as a processing aid for washing of all foods(Food Standards Australia New Zealand. Scientific Assessment of the Public Health and Safety of Poultry Meat in Australia; FSANZ, Ed.; FSANZ: Canberra, Australia, 2005; pp. 1–223.). In EU we have: Scientific Opinion on the evaluation of the safety and efficacy of peroxyacetic acid solutions for reduction of pathogens on poultry carcasses and meat. Moreover a lot of similar article is avaliable in scientific journal: Evaluating best practices for Campylobacter and Salmonella reduction in poultry processing plants
https://doi.org/10.3382/ps/pev328 (6 met plants), The Microbial and Quality Properties of Poultry Carcasses Treated with Peracetic Acid as an Antimicrobial Treatment (https://doi.org/10.3382/ps.2008-00087),
Salmonella and Campylobacter reduction and quality characteristics of poultry carcasses treated with various antimicrobials in a post-chill immersion tank (https://doi.org/10.1016/j.ijfoodmicro.2013.05.016),
Methods of using peracetic acid to treat poultry in a chill tank during processing to increase weight (US patent US20120244261A1) and another.
Authors comment: It would have been great if we could have serotyped all the 79 isolates, however, this could not be completed due to several reasons including cost. We disagree with the comment. In our opinion serotyping is cheap and very common method. Moreover authors detectonly Serotype Schwarzenburg and Kentucky. Based on numerous reports (e.g.
Centers for Disease Control and Prevention, Foodborne Diseases Active Surveillance Network) in US re present: S. Enteritidis, S. Infantis, S.
Heidelberg, S. Typhimurium. Regional differences include higher proportions of serovars Infantis and Typhimurium in the Atlantic and higher proportion of serovar Schwarzengrund in the Southeast were observed. Thus 12 samples from 79 (15,19%) is not sufficient.
Authors comment: Several published studies have already tested the in-vitro efficacy of PAA at different concentrations against both Salmonella and Campylobacter. We agree with this. Whereas, several published studies have already tested PAA in vivo, in meat planst, meat packing plant etc.
Minor revisions:
According to Authors knowledge, there are no such requirements to use a semisolid agar for Salmonella isolation. We agree with the comment. But using semi-solid MSRV agar makes Salmonella isolation easier.
In summ, the manuscript is well written and although the work is not particularly novel, it doesn`t provide original information.
Author Response
We thank the reviewer for the comments/suggestions. Please see the attachment for our response to the comments.
